# Correlates of older adult inpatients' personal care provision to people with functional difficulties in Ghana

Kofi Awuviry-Newton[1]*, Kwadwo Ofori-Dua[2], Abraham Newton[3]

**1** Priority Research Centre for Generational Health and Ageing, School of Medicine and Health, The University of Newcastle, Newcastle, Australia, **2** Department of Sociology and Social Work, Kwame Nkrumah University of Science and Technology, Kumasi, Ghana, **3** Department of Social Studies and Education, University of Education, Winneba, Ghana

* newscous@gmail.com

**Data Availability Statement:** All relevant data are within the manuscript and its Supporting Information files.

**Funding:** This study was funded by The University of Newcastle International Postgraduate Research

## Abstract

### Introduction

Most research depicts older adults as needing personal care, with limited research on older adults' contributions in the lives of others in developing countries like Ghana. The purpose of the study was to examine the personal care provision and socio-demographic correlates of personal care provision by older adult inpatients in Ghana.

### Materials and methods

A hospital-based survey was conducted among 400 consecutively surveyed older adult inpatients at Komfo Anokye Teaching Hospital in southern Ghana. Stata 15 was used to analyse the data. The relationship between personal care and older adult inpatients' socio-demographic characteristics were analysed using the chi-square test. Multivariate logistic regression analysis was employed.

### Results

Overall, 28% of older adult inpatients provided personal care. Participants were mostly females, married or cohabiting, completed at most junior high school, Christians, urban residents, not working and living with their immediate family. Primarily, most participants provided personal care to one person, once a week, and one-hour duration. Nearly three-quarters of participants provided personal care to someone who lives with them. After adjustment, male older adults were 50% less likely to and urban residents were 83% more likely to provide personal care. Being single, separated or divorced was statistically significantly associated with personal care provision, however, were not statistically significant after adjusting for sex and residence. A post hoc analysis testing for interaction revealed no relationship existing between sex and marital status concerning personal care provision (p = 0.106).

Scholarship (UNIPRS) and The University of Newcastle Research Scholarship Central 50:50 (UNRSC50:50). This research was supported by the Australian Research Council Centre of Excellence in Population Ageing Research (project number CE170100005).

**Competing interests:** No author have competing interest

## Conclusion

Female and urban resident older adult inpatients in Ghana are not just passive receivers of care but also provide personal care to others with functional difficulties, independent on age. It further draws attention to the need for policies and programs that can support older adults, particularly females and urban residents, to be productive in the later life.

## Introduction

The older population in Ghana, those aged 60 years or older, is growing rapidly due to decreasing birth rates and increased longevity [1]. Ghana's older population increased more than seven-fold from 213,477 (4.5%) in 1960 to 1,643,381 (6.7%) in 2010 [2]. The percentage of older adults is further expected to increase to 9.6% by 2050 [3]. The growth in Ghana's proportion of older adults has sparked an increase in research depicting older adults as dependent on others for personal care [4–12].

Several authors have defined personal care with different emphasis. Shiel [13] defines it as assistance in the basic activities a person may need to achieve the best out of life. Personal care includes assistance in bathing, grooming, toileting and medication [13]. It also involves instrumental activities such as managing money, household chores shopping, telephone use and preparing meals [9,14], as well as advice, informational, psychological and emotional assistance in dealing with life challenges [15,16]. A lot of research on personal care for older adults in Ghana has been fueled by evidence in Ghana that some older adults experience some levels of functional limitation, participation restrictions in voluntary work and therefore require the support of others [4,10–12,17–19]. Whiles, it is relevant for research to focus on the provision of personal care for older adults in Ghana; a more positive perspective is that many older adults may be productive irrespective of their poor health. These older adults experiencing functional limitations can contribute to the wellbeing of others, particularly those experiencing difficulties meeting their daily needs in society by providing personal care such as assistance with toileting. It is in line with this that a shift in Gerontological research from total dependence to productive ageing gained prominence in the 1980s [20], and it is still relevant today.

Productive ageing refers to a state where older adults use their knowledge, experiences and skills to consistently contribute to their family and community to increase their self-worth and value [21]. It can take varying forms including engaging in either paid or unpaid work; providing care for significant others including family, relatives and friends; volunteering; and pursuing education aimed to enhance their human capital and productivity [21].

In support of this perspective is a study that examined the profiles of caregivers reporting the mean age of caregivers as 61 years [22], depicting that caregivers in Ghana were mostly older adults. These caregivers were providing personal care, emotional and financial assistance to their dependents. In sub-Saharan African countries, studies abound about older adults' involvement in the caring of their grandchildren because of the changing household structure. A study that used data from demographic and health surveys to examine the composition of household structure with older adults from 24 sub-Saharan African countries reported that older adults living with grandchildren with no adult children are a common phenomenon. Although most of the adult children live elsewhere, about 8% of grandchildren have lost one parent. Moreover, in countries with a high prevalence of Human Immunodeficiency Virus (HIV) mortality, older adults are more likely to live with grandchildren who have lost both parents to death [23]. In this family structure, older adults may take the role of parents responsible for childcare and personal care for their grandchildren.

Several reasons exist on why people, including older adults receiving healthcare themselves, may provide personal care to those who need it. According to Schulz [24], people provide care for an expectation of rewards ranging from gaining social approval to fear of disapproval for not providing care. Another explanation is based on altruistic motivation suggesting that the primary aim of caring is sacrificing self-desires to helping the cared-for achieve his or her goal [24,25]. According to Schulz [24], the benefit to the caregiver is just a consequence of providing care. Others may also offer care for reciprocity reasons. People may give care to others as a payback. A longitudinal study conducted in Amsterdam reported that older adult's assistance in childcare for grandchildren for their adult children determine support they receive later in life. This assistance, however, varies according to the gender of their adult children. Older adults who provide care for their sons' children are more likely to receive instrumental and emotional support from their sons. Providing care for their daughters' children does not pay off [26]. Moreover, activity theory suggests that maintenance of activity in old age compensates for role-loss and promotes moral wellness and thus should be encouraged. As a result, providing care to others promote the health of both the cared-for and the caregiver [27].

Studies on the profiles of older adults who assist people with functional difficulties exist in advanced countries. In China, for instance, variation exists among older adults living in urban and rural areas in terms of the amount and kind of assistance they provide. In the study, 41% of older adults living in the urban area provide care to grandchildren compared to 31% by rural-dwellers. In terms of housework, rural older dwellers participate more than urban dwellers (32% vs 42%) [28]. Moreover, older adults, mostly women aged 60–64, used more hours to provide unpaid assistance to either family members or friend [29].

In Ghana, the socio-demographic characteristics of older adults, particularly those hospitalised who provide personal care to those with functional difficulties, the extent of care provision and beneficiaries are unknown. The attributes of caregivers revealed by Sanuade and Boatemaa [22] included those in the middle ages (50–59 years) and population from the community obscuring our understanding of those older adults inpatients who provide personal care. Awareness of the socio-demographic correlates, the extent of care provision and beneficiaries of older adult inpatients' personal care they offer may provide evidence for further research and policymakers for enacting policies and programs that can support older adults in their quest to be productive in their later lives. The present study seeks to fill the gap in knowledge by addressing these research questions: What are the socio-demographic characteristics of older adults inpatients who provide personal care? To what extent do older adults' inpatients provide personal care? Who benefits from the personal care older adults provide? What are the correlates of older adults inpatients who provide personal care?

## Material and methods

### Study area

The survey was conducted in Komfo Anokye Teaching Hospital (KATH) located in Kumasi, the Regional Capital of Ashanti Region with a population of 4,780,249 as of 2010 [2]. Ashanti Region shares boundaries with some regions of Ghana including Brong Ahafo Region to the north, Eastern Region to the east, and Central Region to the south and Western Region to the west [30]. KATH is accessible to a wide range of people from 13 of the 16 administrative regions of Ghana. The hospital receives referrals from these regions. KATH provides medical services to the majority of older adults.

### Study design

Survey descriptive design was employed.

## Study sampling

This study is a portion of the primary author's PhD research conducted at KATH to examine the functional difficulties among older adult inpatients. As part of the objectives of the overall PhD study, participants' involvement in personal care for others in their various homes before their admission to the hospital were also assessed. The population of the present survey was older adults (60 years or older) admitted to the Komfo Anokye Teaching Hospital within the defined sampling period. Random sampling technique was used to select four days per week to recruit participants. All older adults admitted to the hospital at the time of the study were eligible to participate if they met the inclusion criteria. The inclusion criteria were; 1) They must be older adults (60 years and over), 2) They must be admitted to the hospital for any health reason, 3) They must have stayed a minimum of one night, 4) They must not be seriously sick and, 5) They must be willing to participate in the study. On the other hand, older adults were excluded if 1) older adults aged 60 years and over but whose health status is critical and, 2) older adults who did not express willingness to take part in the study [9]. The inclusion criteria were purposeful to determine whether older adults who receive care were also providers of care. However, no prior research identified employing this methodology in the recruitment of participants.

The study used consecutive sampling techniques to recruit individuals who met the eligibility criteria. The study recruited 400 participants during the data collection period. All data was collected at the hospital in the time and day chosen by the participants.

Before recruitment, nurses were asked to seek the consent of older adult inpatients who satisfy the inclusion criteria. The researcher allowed 48 hours for older adult inpatients to give their consent. When a patient agreed, the researcher and a research assistant approached the patient and conducted the structured interviews. The researcher read and translated for participants who could not read and write. All communication was conducted in "Twi", the dominant language of Ghana. The detail of the methodology is explained in Awuviry-Newton et al. [9].

## Data collection

A survey questionnaire was used to collect information from older adults inpatients. The researcher read the questions to participants who could not read to solicit quantitative data. The structured interview was composed of closed-ended questions. Data collection with the participants were completed between August and December 2018.

## Dependent variable: Personal care

Personal care provision was assessed among older adults using a variable with a nominal response category. The question asked was, "Do you regularly provide care or assistance (e.g. bathing, transport) to any other person because of their long-term illness, disability or frailty before you were hospitalised?" The original nominal response categories were; 1) yes, for someone who lives with me, 2) yes, for someone who lives elsewhere and 3) no, I do not provide care. For this analysis the response categories; yes, I do provide care for someone who lives with me; and, yes, I do provide care for someone who lives elsewhere were classified as "yes" and no, I do not provide care was classified as "no". This categorisation is necessary to determine those who provide care and those who do not rather than to identify to whom care is provided as elsewhere [12]. "Yes" was assigned "1" whereas, "No was assigned "0" for multivariate logistic regression.

The extent by which older adults provide personal care has been provided. This variable measuring the extent of personal care provision included; 1) the number of care recipients, 2)

the number of times older adults provide care, and 3) the number of hours used in providing personal care. Moreover, the pictorial description of the beneficiaries of personal care provided by older adults has been presented.

## Independent variables

The variables included in the analysis were age, sex, marital status, education, residence, employment status, living arrangement, and religion. Age was measured as a continuous variable. Marital status was categorised as never married, married or cohabiting, separated or divorced and widowed. For the purpose of analysis, marital status was categorised into three responses as single/separated/divorced, married/cohabiting and widowed. Education was categorised as no education, less than primary school, primary school, secondary school, high school, and college/pre-university, undergraduate and postgraduate degree. Education response categories were further categorised as no education, at maximum junior high completed and at least senior high completed. A residence was measured as living in either rural or urban areas. Concerning the living arrangement, this variable was initially measured as "alone", "with husband/wife", "with children/with husband/wife and children", and "extended family". The responses for living arrangement were then categorised as "alone" for alone, "living as a couple" for living with husband/wife, and "with partner and children" for with children/with husband/wife and children, and extended family. Employment status was measured as "currently working" and "currently not working".

## Data analysis

All analysis was facilitated by Stata 15. Descriptive statistics were used to describe essential variables in the study population. Chi-square test was used to determine the relationship between independent variables and the dependent variable. Univariate and multivariable logistic regression was performed to assess any significant relationship between independent variables and the dependent variable. Crude and adjusted odds ratios were used to ascertain any associations between the dependent and independent variables using a 95% confidence interval to determine the level of significance. Any significant association was determined at a p-value of less than 0.05 in the multivariable logistic regression model to control for potential confounding variables. Logistic regression analysis was conducted to determine the correlates of personal care provision among older adults inpatients in Ghana.

## Ethical consideration

Ethical approval for this analysis was obtained from The University of Newcastle (H-2018-0163), New South Wales, Australia and Kwame Nkrumah University of Science and Technology (CHRPE/AP/112/18), in Ghana in keeping with the Declaration of Helsinki. Written Informed consent was obtained from the director of Komfo Anokye Teaching Hospital and study participants. Anonymity and confidentiality were ensured.

## Results

Twenty-eight per cent of participants provided care. The average age of the 400 participants was approximately 71 years, and most were females (51.0%). Majority of participants were married or cohabiting (53.0%), completed at least junior high school (52.3%), mostly Christians (82.8%) and living in the rural area (56.8%). Moreover, most participants were living with their immediate family (62.5%) and were not working (61.0%) (see Table 1).

**Table 1. Socio-demographic characteristics of participants.**

| Demographic Characteristics (N = 400) | N (%) |
|---|---|
| **Age** *(mean, SD)* | 71.3±8.42 |
| **Sex** | |
| Male | 196 (49.0 |
| Female | 204 (51.0) |
| **Marital status** | |
| Single/separated/divorced | 58 (14.5) |
| Currently married/cohabiting | 212 (53.0) |
| Widowed | 130 (32.5) |
| **Education** | |
| No education | 128 (32.0) |
| At maximum junior high completed | 209 (52.3) |
| At least senior high completed | 63 (15.8) |
| **Religion** | |
| None | 27 (6.75) |
| Christianity | 331 (82.8) |
| Islam | 42 (10.0) |
| **Residence** | |
| Rural | 227 (56.8) |
| Urban | 173 (43.3) |
| **Living arrangement** | |
| Alone | 50 (12.5) |
| With couple | 100 (25.0) |
| With couple and children | 250 (62.5) |
| **Employment status** | |
| Currently working | 156 (39.0) |
| Currently not working | 244 (61.0) |
| **Provide personal care?** | |
| Yes | 112 (28) |
| No | 288 (72) |

## Older adult inpatients who provide personal care

The univariate analysis shows that the average age of participants who provide personal care was approximately 72 years. Participants who provided personal care were mostly females (60.7%), and married or cohabiting (50%). Furthermore, more than half had completed at most junior high (58.0%), mostly Christians (80.4%), living in the urban areas (55.4%), with the majority living with their immediate family (61.6%) and mostly not working (66.1%) (see Table 2).

## The extent by which older adult inpatients provide personal care to others

Most of the personal care was provided to one person (15.3%), often offered once a week, once every few weeks (12.0%) and usually lasted for about an hour (6.75%) (see Table 3).

## Who receives the personal care older adult inpatients provide?

Nearly three-quarters of older adults offers personal care to someone who lives with them (73.2%) (see Fig 1).

**Table 2. Univariate analysis of older adult inpatients who provide personal care.**

| Characteristics | Personal Care | | | |
|---|---|---|---|---|
| | Total N (%) | Yes N (%) | No N (%) | p-value |
| **Age** | 71.3±8.42 | 71.9±8.91 | 71.0±8.23 | 0.370 |
| **Sex** | | | | 0.016 |
| Male | 196 (100) | 44 (22.5) | 152 (77.6) | |
| Female | 204 (100) | 68 (33.3) | 136 (66.7) | |
| **Marital status** | | | | <0.01 |
| Single/separated/divorced | 58 (100) | 26 (44.8) | 32 (55.2) | |
| Married/cohabiting | 212 (100) | 56 (26.4) | 156 (73.6) | |
| Widowed | 130 (100) | 30 (23.1) | 100 (76.9) | |
| **Education** | | | | 0.244 |
| No education | 128 (100) | 34 (26.6) | 94 (73.4) | |
| At most junior high completed | 209 (100) | 65 (31.1) | 144 (68.9) | |
| At least senior high completed | 63 (100) | 13 (20.6) | 50 (79.4) | |
| **Religion** | | | | 0.692 |
| None | 27 (100) | 8 (29.6) | 19 (70.4) | |
| Christianity | 331 (100) | 90 (27.2) | 241 (72.8) | |
| Islam | 42 (100) | 14 (33.3) | 28 (66.7) | |
| **Residence** | | | | <0.01 |
| Rural | 227 (100) | 50 (22.0) | 177 (78.0) | |
| Urban | 173 (100) | 62 (35.8) | 111 (64.2) | |
| **Living arrangement** | | | | 0.850 |
| Alone | 50 (100) | 13 (26.0) | 37 (74.0) | |
| With couple | 100 (100) | 30 (30.0) | 70 (70.0) | |
| With couple and children | 250 (100) | 69 (27.0) | 181 (72.4) | |
| **Employment status** | | | | 0.195 |
| Currently working | 156 (100) | 38 (24.4) | 118 (75.6) | |
| Currently not working | 244 (100) | 74 (30.3) | 170 (69.7) | |

Significant at *p-value < 0.05; **p-value < 0.01.

## Correlates of personal care provision by older adults inpatients

After adjustment, the multivariate analysis revealed that male older adults were 50% less likely to provide personal care compared to female older adults (AOR (95% C.I): 0.50 (0.31, 0.81)). Compared to the unadjusted odds ratio, the odds ratio reduced by 8%, when adjusted for marital status and residential location. Being single, separated or divorced was statistically significantly associated with personal care provision (COR (95% C.I): 2.26 (1.24, 4.13), however, became dependently statistically not significant after adjusting for sex and residence. Concerning the widow marital status, it was independently not significantly related to the personal care provision before and after adjustment (see Table 4).

Living in an urban area was independently statistically significantly associated with personal care provision. Being an urban resident was 83% more likely to provide personal care to someone with functional difficulties (AOR (95% C.I): 1.83 (1.16, 2.89)) compared to rural dwellers. There was a 15% reduction in the odds ratio when adjusted for sex and marital status. A post hoc analysis to test for interaction revealed no relationship existing between sex and marital status related to personal care provision (p = 0.106).

**Table 3. Extent of which older adults provides personal care.**

| The extent of personal care provision (N = 112) | Number (N) | Percentage (%) |
|---|---|---|
| **Number of people you regularly care** | | |
| **None** | 288 | 72.0 |
| One person | 61 | 15.3 |
| Two people | 32 | 8.00 |
| More than two people | 19 | 4.75 |
| **Often assistance is provided** | | |
| **None** | 288 | 72.0 |
| Every day, several times a week | 30 | 7.50 |
| Once a week, once every few weeks | 48 | 12.0 |
| Less often | 34 | 8.50 |
| **Number of times spent on care** | | |
| **None** | 288 | 72.0 |
| All day and night | 24 | 6.00 |
| All night | 23 | 5.75 |
| All day | 18 | 4.50 |
| Several hours | 20 | 5.00 |
| About an hour | 27 | 6.75 |

## Discussion

Our findings indicate that older adults inpatients contribute to the wellbeing of others through **the** provision of personal care. The finding by this current study that older adults provide personal care corroborates with the finding of a study conducted in the UK that older adults

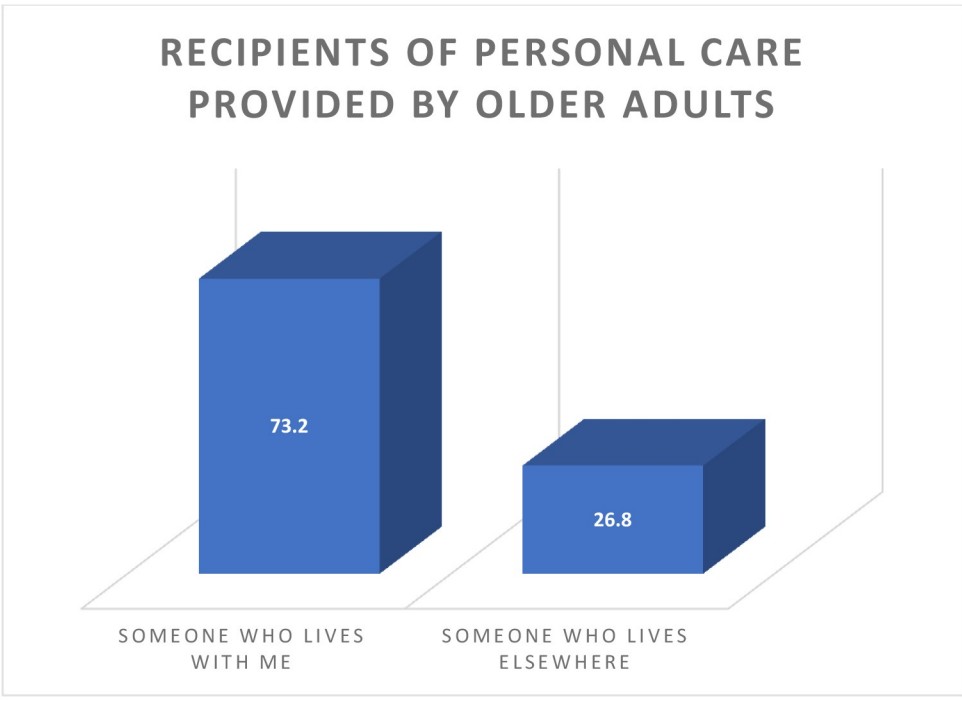

**Fig 1. Who receives the personal care older adults provide?**

**Table 4. Multivariate analysis of the correlates of personal care provided by older adult inpatients.**

| Variables | COR 95%CI | AOR 95%CI |
|---|---|---|
| **Sex** | | |
| Male | 0.58 (0.37, 0.90)* | 0.50 (0.31, 0.81) ** |
| Female | 1 | 1 |
| **Marital status** | | |
| Single/separated/divorced | 2.26 (1.24, 4.13)** | 1.83 (0.98, 3.41) |
| Currently Married/cohabiting | 1 | 1 |
| Widowed | 0.84 (0.50, 1.39) | 0.65 (0.38, 1.12) |
| **Residence** | | |
| Rural | 1 | 1 |
| Urban | 1.98 (1.27, 3.08)** | 1.83 (1.16, 1.2.89)** |

Significant at *p-value < 0.05

**p-value < 0.01.

provide a huge contribution to society [31]. This affirmation concludes that older adults are active providers of care irrespective of their poor health and therefore, government institutions, friends, or families should nurture this potential. The smaller proportion of the older adults contributions compared to the study in the UK could be that the present study sampled older adult inpatients. In contrast, the UK study sampled community-dwelling older adults. After adjustment and controlling for confounding, female older adults and urban-dwelling older adults are more likely to provide personal care to someone who needed it.

The finding that female older adults provide personal care could be understood in the cultural context throughout the life course in Ghanaian society. Culturally and traditionally, women are more engaged in care, including personal care whereas men are breadwinners and decision-makers in the family [6]. In the old age, women find their identity unchanged and so are more inclined into caring for people needing care compared to older men. This finding corroborates other findings that older women provide more assistance to family members or friend [29], which may include personal care. A significant concern is how people with functional difficulties will be cared for, in this era where women in Ghana are attaining higher education and migrating following urbanization and modernization. We recommend that health and welfare service that can support older adults' especially older women to be productive and contribute to wellbeing of others as far as possible will be helpful.

The previous study conducted in China reported that most of the older adults who provide care to grandchildren are urban dwellers [28]. Consistent with this previous study, the current study report that older adult urban dwellers are 16% more likely to provide personal care than older adults' rural dwellers. This practice is common because of the existence of at least three-generational households in Ghana [32]. Most of the time, adult children in the urban cities invite their parents to live with them in the cities and by so doing; older adults may be providing personal care to a person with functional difficulties or providing childcare.

Age was not significantly associated with personal care provision in this study. This finding is contrary to previous mainstream studies which report an association between age and personal care [27,33]. The variation in findings may be attributed to the study settings. In this current study, older adult inpatients were recruited from a hospital setting, where all participants irrespective of the age may have been experiencing poor health. However, the current study asked participants to reflect on care provision at home, before they were admitted to the hospital. Further research recruiting community-dwelling older adults in Ghana on personal care provision is relevant to understanding any variation that may be existing. This research will

help in understanding the health and social needs of these older adults who provide care and support.

As evident in the current study, participants provided personal care to one person, once a week, and mostly on-hour duration. Although participants mostly offer care for one hour, the percentages of the length of categories of care provision are quite similar, implying that older adults could provide care to any length. Moreover, the finding that most participants provided personal care to someone who lives with them infers that participants provide most of the personal care to either their spouse, own children or family members who may stay with participants due to their illness. The number of time participants put in caregiving and how often demonstrate their interest in enhancing the wellbeing of others. Participants may also be improving their emotional wellbeing and happiness as posited by the activity theory [27]. Due to the effort older adults put in to contribute to the wellbeing of others, we suggest that there should be community support for older adults in Ghana to nurture their potentials, especially when older adults inpatients are discharged from the hospital setting.

The findings from this current study have implications for research, policy and practice. In terms of research, the health and social care needs of older adults who provide personal care should be explored. Research is also needed to understand older adults 'contributions in volunteering services and child care to complement the existing evidence in this study. Moreover, more research needs to explore how the social and physical environment can nurture the contributions of older adults in the Ghanaian setting. This understanding is essential to facilitate practices to ensure productive ageing. Concerning policy, the existing policy on older adults developed in Ghana in 2010 [34] should be implemented to help Ghana amass the unique contributions of older adults in various avenues. Specifically, the policy seeks to reduce poverty and improve healthcare among older adults; however, evidence in Ghana shows that some older adults do not have good health and feel neglect owing to the deprivation from their own family as well as the government institutions [10]. In terms of practice, health and social care professionals should promote older adults ability to be productive through healthcare and counselling. The finding in this study offers social workers the opportunity to capitalise on older adults' contribution to increase older adults' economic wellbeing to reduce the impact of poverty on the overall functioning of older adults.

This study has some strengths that need to be acknowledged. First, this study using older adult inpatients from the hospital and considering the finding that older adults contribute to the wellbeing of others echoes our understanding that older adults are not just passive receivers but also caregivers. Sampling participants from one hospital may have missed other participants who did not come to the hospital; however, the hospital receives patients from about 13 regions of the 16 regions in Ghana. Moreover, probability sampling was employed to recruit participants and may have reduced some bias. Another potential limitation for this study is that though the activity theory applied in this study is of the assumption that the healthy older adults provide care; however, the current study sampled hospitalised patients. Although, the literature search did not identify any study on the health status of older adults' caregivers, the current study provides an opportunity for new research exploring the health status of older adults who care for others. A study sampling a large sample size of older adults' caregivers will help to understand their health and social care needs.

## Conclusion

Though a small proportion of older adults receiving health care from the hospital setting provide personal care, the findings in this study demonstrate that older adults in the community may be resourceful to the community and their family members due to their relatively better

health. Moreover, the study's findings draw attention to the need to provide financial and health support to older adults receiving healthcare who do not provide personal care to enable them to be productive. In this period where Ghana's population is reportedly growing old, the proportion and number of people aged 60 years or older will increase. These populations are valuable resources to their families, communities and the country as a whole. The health and social care needs of all older adults should be ensured so that older adults can remain productive as far as possible. The findings have drawn attention to the policy and research initiatives to facilitate productive ageing.

## Supporting information

**S1 Appendix.**
(DOCX)

## Acknowledgments

Thanks to the participants for their time in participating in the survey.

## Author Contributions

**Conceptualization:** Kofi Awuviry-Newton.

**Formal analysis:** Kofi Awuviry-Newton.

**Methodology:** Kofi Awuviry-Newton, Kwadwo Ofori-Dua, Abraham Newton.

**Software:** Kofi Awuviry-Newton.

**Supervision:** Kwadwo Ofori-Dua, Abraham Newton.

**Visualization:** Kofi Awuviry-Newton.

**Writing – original draft:** Kofi Awuviry-Newton.

**Writing – review & editing:** Kofi Awuviry-Newton, Kwadwo Ofori-Dua, Abraham Newton.

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
