## [Decision Letter · Decision Letter 0]

20 May 2020

PONE-D-19-28535

Socio-demographic Predictors of Older Adults’ Personal Care Provision to People with Functional Difficulties in Ghana.

PLOS ONE

Dear Dr Awuviry-Newton,

Thank you for submitting your manuscript to PLOS ONE. After careful consideration, we feel that it has merit but does not fully meet PLOS ONE’s publication criteria as it currently stands. Therefore, we invite you to submit a revised version of the manuscript that addresses the points raised during the review process.

Please accept my apologies for the long delay in the review of this paper.  It has been difficult to find reviewers.  In order not to prolong the process, I have undertaken a review of the paper myself, prior to reading the single review of the other reviewer.  The recommendations of both reviews were the same: Major Revision.  Although the two reviews make different suggestions, they are all valid and complementary.  In particular: Please edit for scientific English.  I have indicated page/line numbers that need attention.  Please justify your sample design by tailoring your literature to focus on the health status of older caregivers. Please report the row percentages in your Table 2 so they do not simply reflect the proportion in the sample.  Please develop some policy recommendations relevant for improving conditions for this group.

We would appreciate receiving your revised manuscript by Jul 04 2020 11:59PM. To enhance the reproducibility of your results, we recommend that if applicable you deposit your laboratory protocols in protocols.io, where a protocol can be assigned its own identifier (DOI) such that it can be cited independently in the future. For instructions see: http://journals.plos.org/plosone/s/submission-guidelines#loc-laboratory-protocols

We look forward to receiving your revised manuscript.

Kind regards,

Ellen Idler

Academic Editor

PLOS ONE

Additional Editor Comments (if provided):

Because of the difficulty in finding a second reviewer for this paper, I am undertaking a review myself. (See below). I performed this review without previously reading the comments of Reviewer 1.

In my role as Editor, I agree with the suggestions and critique given by Reviewer 1, and would urge the authors to attend to the suggestions of both reviewers.

Review of “Socio-Demographic Predictors of Older Adults’ Personal Care Provision to People with Functional Difficulties in Ghana”

Manuscript ID: PONE-D-19-28535

This paper reports results from a survey of consecutive inpatients admitted to a hospital in Ghana, on the topic of the patients’ caregiving for others (N=400). The purpose is descriptive: what are the characteristics of older adult caregivers in Ghana? Results from multivariate logistic regression showed that females were 50% more likely to provide caregiving than males, and that urban dwellers were 83% more likely to be caregivers. Strengths of the manuscript are the novelty of the topic in an African country where it has not been well-studied, and the relatively large sample size of older Ghanaians.

1. There are a large number of grammatical errors which should be edited for scientific English before the paper is accepted. Common issues are definite and indefinite articles and subject-verb agreement. I will list page and line numbers here:

Page Line

1 10, 16

2 1, 5, 6, 7

3 7, 17, 20, 22, 23

4 3, 15

5 12, 17

6 17, 20

7 5, 22

8 4, 8, 18

9 9, 17, 18, 20

10 11

15 7 in paragraph

16 2 in paragraph

17 1, 3, 8

18 2, 4-6 sentence fragment, 9, 16, 19

19 1

2. Please do not use the word “predictors” when describing cross-sectional results. “Correlates” is more precise.

3. There is an important issue of selection bias in the sample. Those who were sampled were aged 60 and older, and were being admitted to hospital. The subject of the study is the caregiving role of these admitted patients, so the ill are being asked about a role usually occupied by a healthy person (suggested by activity theory, mentioned on p.5). This potential limitation is raised in the Discussion, so the authors are aware of it, but they could capitalize on it better. On p. 7 it says “The inclusion criteria were purposeful to determine whether older adults who receive care were also providers of care.” I think the literature review prior to this could have been better targeted to focus on the health of older caregivers. If earlier studies did assess the health of older caregivers that should be included; if they did not, this is an opportunity for new research.

4. It would be helpful to include the length of time the interview took (p.8).

5. On p. 9 the description of the “residence” variable shows the original response categories and the revised categories, which eliminate “living with extended family” – and yet the importance of caregiving in multigeneration households is discussed on p. 17 in the discussion.

6. Table 2 shows the cross-tabulation of caregiving by demographic characteristics. Unfortunately, the percentages reported are column %, when they should be row %. We want to know what percent of females are caregivers to compared to the % of males, not what % of caregivers are male and female. Because there are more females in the sample, there will be more female caregivers, almost by definition. This is the same for all categorical variables in the table.

7. Please provide more information on Table 4. The first column is for unadjusted odds ratios, and the second is adjusted – for everything in the table, or for other variables as well?

8. The second sentence of the Discussion (p.16) says that caregivers are more likely to have completed senior high school. But education is not included in Table 4, and in Table 2 the most educated group is by far the smallest, and by the column % the least likely to be providing care. Please explain.

9. The characterization of Table 3 in the Discussion seems a bit misleading. It is true that the largest percent provides care for only one hour, but the percentages for all the categories of the length of providing care are quite similar.

2. Please state in your methods section the participant recruitment date.

3. Please provide the full name of the ethics committee which approved this study in your methods section and on the online submission form.

4. Please include additional information regarding the survey or questionnaire used in the study and ensure that you have provided sufficient details that others could replicate the analyses. For instance, if you developed a questionnaire as part of this study and it is not under a copyright more restrictive than CC-BY, please include a copy, in both the original language and English, as Supporting Information.

5. Please ensure that your references are formatted according to the PLOS ONE submission guidelines https://journals.plos.org/plosone/s/submission-guidelines#loc-references. In particular, please note that references should be listed at the end of the manuscript and numbered in the order that they appear in the text. In the text, please cite the reference number in square brackets.

6. Please specify in the ethics statement in the Methods section and online submission information whether you obtained informed verbal or informed written consent from the participants included in the study. If consent was verbal, please amend your current ethics statement to explain 1) why written consent was not obtained, 2) how you recorded/documented participant consent, 3) whether your ethics committee approved this consent procedure.

7. In your Data Availability statement, you have not specified where the minimal data set underlying the results described in your manuscript can be found. PLOS defines a study's minimal data set as the underlying data used to reach the conclusions drawn in the manuscript and any additional data required to replicate the reported study findings in their entirety. All PLOS journals require that the minimal data set be made fully available. For more information about our data policy, please see http://journals.plos.org/plosone/s/data-availability.

8. Thank you for stating the following in the Acknowledgments Section of your manuscript:

"This study was funded by The University of Newcastle International Postgraduate Research

Scholarship (UNIPRS) and The University of Newcastle Research Scholarship Central 50:50

(UNRSC50:50).

This research was supported by the Australian Research Council Centre of Excellence in

Population Ageing Research (project number CE170100005)."

"No specific funding received"

Reviewers' comments:

Reviewer's Responses to Questions

**Comments to the Author**

1. Is the manuscript technically sound, and do the data support the conclusions?

Reviewer #1: Partly

2. Has the statistical analysis been performed appropriately and rigorously? 

Reviewer #1: Yes

3. Have the authors made all data underlying the findings in their manuscript fully available?

Reviewer #1: Yes

4. Is the manuscript presented in an intelligible fashion and written in standard English?

Reviewer #1: Yes

5. Review Comments to the Author

Reviewer #1: This paper is well written and the data analysis undertaken is sound enough. The paper achieves what it sets out to do, which is to provide socio-demographic characteristic of older adult providers of personal care in Ghana and the extent of care provided.

The strengths of the paper are that the authors review the relevant bodies of literature adequately, including providing definitions of care and setting the context for understanding the care landscape in Ghana. The authors also undertake some comparative analysis in looking at other cases e.g. China to compare to the Ghanaian case. In terms of basic providing empirical data on older adult care givers in Ghana, the paper achieved its stated aims.

Weaknesses of the paper:

While the paper provides a basic empirical overview of the demographics of older adult care workers, the paper does not make any major theoretical contributions. In other words, it would be helpful for the authors to show what the Ghanaian case study adds to the larger discussions on ageing populations and care. There is a global literature on this that the authors would do well to address. Also the authors allude to the fact that the purpose of the paper was to inform policy makers to better able to support older adults and care in Ghana. However the authors do not offer any specific policy initiatives or suggestions. It would be helpful for the authors to address these issues for a stronger paper. These issues should be addressed before the paper is eligible for print.

Finally, the paper exhibited some minor errors in writing and expression that should be corrected before the paper is brought to print:

1) The abstract is unclear and misleading. The statement the socio-demogrpahic predictors of personal care is unclear. I would suggest the authors amend to read the socio-demographic characteristics of older adult care givers in Ghana

2) Writing errors:

Section study sampling

line 3 participants' participation is clumsy - rephrase

line 4 These study should read "This study"

The statement in line 7-8 "Four days per week was determined through random sampling technique" is unclear. It seems that participant selection could be done through random sampling not the number of days for sampling. This statement should be restated for clarity

6. PLOS authors have the option to publish the peer review history of their article (what does this mean?). If published, this will include your full peer review and any attached files.

Reviewer #1: No

---

## [Author Response · Author response to Decision Letter 0]

4 Jun 2020

Dear Dr Ellen Idler, 

We are very grateful for the privilege given us to revise the manuscript. In this letter, we have responded to the reviewers’ and your comments. 

Response to reviewers 

Response to reviewers: Socio-Demographic Predictors of Older Adults’ Personal Care Provision to People with Functional Difficulties in Ghana”. Manuscript ID: PONE-D-19-28535

1.Because of the difficulty in finding a second reviewer for this paper, I am undertaking a review myself. (See below). I performed this review without previously reading the comments of Reviewer 1.

Authors appreciate the effort the editor has taken to review this manuscript. We acknowledge that we have addressed all the comments raised.

2.In my role as Editor, I agree with the suggestions and critique given by Reviewer 1, and would urge the authors to attend to the suggestions of both reviewers.

Thank you. We have responded to all the comments raised by both reviewers.

3. There are a large number of grammatical errors, which should be edited for scientific English before the paper is accepted. Common issues are definite and indefinite articles and subject-verb agreement. I will list page and line numbers here:

Page Line

1 10, 16

2 1, 5, 6, 7

3 7, 17, 20, 22, 23

4 3, 15

5 12, 17

6 17, 20

7 5, 22

8 4, 8, 18

9 9, 17, 18, 20

10 11

15 7 in paragraph

16 2 in paragraph

17 1, 3, 8

18 2, 4-6 sentence fragment, 9, 16, 19

19 1

Authors appreciate this in-depth suggestions. We have proofread the manuscript for subject-verb agreement and other typographical errors. 

4. Please do not use the word “predictors” when describing cross-sectional results. “Correlates” is more precise.

We have corrected the whole manuscript per this suggestion. We have replaced “predictors” with “correlates”, including the title.

5. There is an important issue of selection bias in the sample. Those who were sampled were aged 60 and older, and were being admitted to hospital. The subject of the study is the caregiving role of these admitted patients, so the ill are being asked about a role usually occupied by a healthy person (suggested by activity theory, mentioned on p.5). This potential limitation is raised in the Discussion, so the authors are aware of it, but they could capitalize on it better. 

Thank you for the in-depth comments and analysis. However, we have elaborated on this limitation in the discussion. We have added that “Another potential limitation for this study is that though the activity theory applied in this study is of the assumption that the healthy older adults provide care, however the current study sampled hospitalised patients. Though, the literature search did not identify any study on the health status of older adults caregivers, the current study provides an opportunity for new research exploring the health status of older adults who care for others. This study can sampled a large sample size of older adults’ caregivers to understand their health and social care needs.” This portion can be found under “discussion” on page 19.

6. On p. 7 it says “The inclusion criteria were purposeful to determine whether older adults who receive care were also providers of care.” I think the literature review prior to this could have been better targeted to focus on the health of older caregivers. If earlier studies did assess the health of older caregivers that should be included; if they did not, this is an opportunity for new research

The literature review conducted did not identify articles reporting on the health status of older adults caregivers and in view of this we have acknowledged the need for further research on the health status of older adults caregiver particularly in the developing countries including Ghana. We have added the sentence “although no prior research identified employing this methodology in the recruitment of participants” under “study sampling” on page 7.

7. It would be helpful to include the length of time the interview took (p.8).

Thank you for this comment. We have included the statement “Data collection with the participants were completed between August and December 2018.” We have included this sentence under “Data collection” on page 8.

8. On p. 9 the description of the “residence” variable shows the original response categories and the revised categories, which eliminate “living with extended family” – and yet the importance of caregiving in multigenerational households is discussed on p. 17 in the discussion.

Thank you for this. The residence variable was measured using two dichotomous category (rural and urban). This dichotomous response category was not changed. Concerning the living arrangement, the initial response category included “living in extended family”, however, it was added to the category “with couple and children” because only two participants reported “living in the extended family”. I have updated this under “independent variables”.

9. Table 2 shows the cross-tabulation of caregiving by demographic characteristics. Unfortunately, the percentages reported are column %, when they should be row %. We want to know what percent of females are caregivers to compare to the % of males, not what % of caregivers are male and female. Because there are more females in the sample, there will be more female caregivers, almost by definition. This is the same for all categorical variables in the table.

Thanks for this in-depth comments. We really appreciate this in-depth contribution. I have updated the Table 2 per the comments.

10. Please provide more information on Table 4. The first column is for unadjusted odds ratios, and the second is adjusted – for everything in the table, or for other variables as well?

I have provided more details in the Table 4 by looking at the relationship between the adjusted and unadjusted odds ratios. For instance, I have included that “Compared to the unadjusted odds ratio, the odds ratio reduced by 8%, when adjusted for marital status and residential location”. Moreover, I have included that “Concerning the widow marital status, it was independently not significantly related to the personal care provision before and after adjustment”. These have been added to information on Table 4 on page 14-15.

11. The second sentence of the Discussion (p.16) says that caregivers are more likely to have completed senior high school. But education is not included in Table 4, and in Table 2 the most educated group is by far the smallest, and by the column % the least likely to be providing care. Please explain.

Since education was not significant in the adjusted multivariate analysis, I agree with the reviewer that this is misleading. In view of this, I have removed education in the discussion. In the Table 2, I have adjusted by computing the row % instead of the column %.

12. The characterization of Table 3 in the Discussion seems a bit misleading. It is true that the largest percent provides care for only one hour, but the percentages for all the categories of the length of providing care are quite similar.

Thank you for raising this important issue. Authors have addressed the comment by adding that “Although participants mostly provide care for one hour, the percentages of length of categories of care provision were quite similar, implying that older adults could provide care to any length.” I have added this explanation in the “discussion” on page 17.

Journal requirement:

13. Please ensure that your manuscript meets PLOS ONE's style requirements, including those for file naming. The PLOS ONE style templates can be found at http://www.plosone.org/attachments/PLOSOne_formatting_sample_main_body.pdf and http://www.plosone.org/attachments/PLOSOne_formatting_sample_title_authors_affiliations.pdf

Thank you for these requirements. I have updated the templates per the journal requirement.

14. Please state in your methods section the participant recruitment date.

I have stated in the section the participants recruitment date. Specifically, I have included that “Data collection with the participants were completed between August and December 2018.” These can be found under “data collection” on page 8.

15. Please provide the full name of the ethics committee, which approved this study in your methods section and on the online submission form.

The name of the ethics committees have been provided under “Ethical consideration” on page 10. The detail is provided as “Ethical approval for this analysis was obtained from The University of Newcastle (H-2018-0163), New South Wales, Australia and Kwame Nkrumah University of Science and Technology (CHRPE/AP/112/18), in Ghana in keeping with the Declaration of Helsinki. Informed consent was obtained from the director of Komfo Anokye Teaching Hospital and study participants. Anonymity and confidentiality was ensured.” 

16. Please include additional information regarding the survey or questionnaire used in the study and ensure that you have provided sufficient details that others could replicate the analyses. For instance, if you developed a questionnaire as part of this study and it is not under a copyright more restrictive than CC-BY, please include a copy, in both the original language and English, as Supporting Information.

I have provided a copy of the questionnaire as a supporting information.

17. Please ensure that your references are formatted according to the PLOS ONE submission guidelines https://journals.plos.org/plosone/s/submission-guidelines#loc-references. In particular, please note that references should be listed at the end of the manuscript and numbered in the order that they appear in the text. In the text, please cite the reference number in square brackets

I have updated the manuscript accordingly, by using the correct referencing format.

18. Please specify in the ethics statement in the Methods section and online submission information whether you obtained informed verbal or informed written consent from the participants included in the study. If consent was verbal, please amend your current ethics statement to explain 1) why written consent was not obtained, 2) how you recorded/documented participant consent, 3) whether your ethics committee approved this consent procedure.

Thank you for this in-depth comment. I have specified in the manuscript under “ethical consideration” on page 10 that “Written Informed consent was obtained from the director of Komfo Anokye Teaching Hospital and study participants. Anonymity and confidentiality was ensured”.

19. In your Data Availability statement, you have not specified where the minimal data set underlying the results described in your manuscript can be found. PLOS defines a study's minimal data set as the underlying data used to reach the conclusions drawn in the manuscript and any additional data required to replicate the reported study findings in their entirety. All PLOS journals require that the minimal data set be made fully available. For more information about our data policy, please see http://journals.plos.org/plosone/s/data-availability.

Thank you for this. Per The University of Newcastle ethical requirement, I am not required to share the minimal data set. In view of this, I have stated under the “Data Availability statement” that I am not including data set due to the university’s policy on privacy. I have changed the status of the data availability as “no-some restrictions will apply” to align with the ethics committees policy.

20. Thank you for stating the following in the Acknowledgments Section of your manuscript:

"This study was funded by The University of Newcastle International Postgraduate Research Scholarship (UNIPRS) and The University of Newcastle Research Scholarship Central 50:50 (UNRSC50:50). This research was supported by the Australian Research Council Centre of Excellence in Population Ageing Research (project number CE170100005)."

"No specific funding received"

Thank you for this emphasis. Authors have removed the funding statement from the research.

Reviewer #1

1. This paper is well written and the data analysis undertaken is sound enough. The paper achieves what it sets out to do, which is to provide socio-demographic characteristic of older adult providers of personal care in Ghana and the extent of care provided. The strengths of the paper are that the authors review the relevant bodies of literature adequately, including providing definitions of care and setting the context for understanding the care landscape in Ghana. The authors also undertake some comparative analysis in looking at other cases e.g. China to compare to the Ghanaian case. In terms of basic providing empirical data on older adult caregivers in Ghana, the paper achieved its stated aims.

Authors appreciate reviewer’s acknowledgement of the quality of the paper. We are emphasising that authors have addressed all the comments raised. 

2. While the paper provides a basic empirical overview of the demographics of older adult care workers, the paper does not make any major theoretical contributions. In other words, it would be helpful for the authors to show what the Ghanaian case study adds to the larger discussions on ageing populations and care. There is a global literature on this that the authors would do well to address. 

Thank you for this concern. From the literature, it was hard identifying literature; however, I had access to one literature, which I have included in this discussion. I have updated the discussion by adding some findings from relevant literature. For instance, I have added this section “The finding by this current study that older adults provide personal care corroborate with finding of a study conducted in the UK that older adults provide a huge contribution to society (Fenton & Draper, 2014). This affirmation concludes that older adults are active providers of care and support and therefore government institutions, friends or families should nurture these potential.”

3. Also the authors allude to the fact that the purpose of the paper was to inform policy makers to better able to support older adults and care in Ghana. However, the authors do not offer any specific policy initiatives or suggestions. It would be helpful for the authors to address these issues for a stronger paper. These issues should be addressed before the paper is eligible for print.

I have provided specific details on the policy implications given the findings found in the study “Concerning policy, the existing policy on older adult developed in Ghana in 2010 (Government of Ghana, 2010) should be implemented to help Ghana amass the unique contributions of older adults in various avenues. Specifically, the policy seeks to reduce poverty and improve healthcare accessibility among older adults, however, evidence in Ghana shows that some older adults do not have good health and feel neglect owing to the deprivation from their own family as well as the government institutions (Awuviry-Newton, Nkansah, & Ofori-Dua, 2020).

4. Finally, the paper exhibited some minor errors in writing and expression that should be corrected before the paper is brought to print: 

1) The abstract is unclear and misleading. The statement the socio-demographic predictors of personal care is unclear. I would suggest the authors amend to read the socio-demographic characteristics of older adult caregivers in Ghana

Thank you for addressing this important point. We have addressed this misleading statement by using “correlate”.

5. 2) Writing errors:

Section study sampling

line 3 participants' participation is clumsy - rephrase

Thanks to the authors for highlighting this error. Authors have rephrase it as “As part of the objectives of the overall PhD study, participants’ involvement in personal care for others in their various homes before their admission to the hospital were also assessed”

6. Line 4 These study should read "This study"

Thank you for this. I have corrected the error as “this study”. 

7. The statement in line 7-8 "Four days per week was determined through random sampling technique" is unclear. It seems that participant selection could be done through random sampling not the number of days for sampling. This statement should be restated for clarity.

Authors agree with the reviewer. Authors have corrected the expression as “Random sampling technique was used to select four days per week to recruit participants”.

---

## [Decision Letter · Decision Letter 1]

31 Jul 2020

PONE-D-19-28535R1

Socio-demographic correlates of older adults’ personal care provision to people with functional difficulties in Ghana.

PLOS ONE

Dear Dr. Awuviry-Newton,

Thank you for submitting your manuscript to PLOS ONE. After careful consideration, we feel that it has merit but does not fully meet PLOS ONE’s publication criteria as it currently stands. Therefore, we invite you to submit a revised version of the manuscript that addresses the points raised during the review process.

The comments from previous reviews have mostly been attended to, but there are some remaining issues.  Please edit the paper for a more appropriate characterization of the findings.  

We look forward to receiving your revised manuscript.

Kind regards,

Ellen L. Idler

Academic Editor

PLOS ONE

Journal Requirements:

Additional Editor Comments (if provided):

The author has addressed most of the suggestions for revision made in the two reviews. There are a few remaining issues, however, that can be addressed in a Minor Revision.

1. Please reframe the research question in the Abstract and elsewhere. As stated "The purpose of the research is to determine the extent of care provision and particularly examine the socio-demographic correlates of personal care provision by older adults in Ghana" -- this research question is posed as descriptive of the extent of personal care provision by ALL older adults in Ghana (the population-level) but the data come from a highly selected sample of patients, from which no generalizations can be made to any larger group.

2. Please provide a more realistic assessment of the extent of care provision. Only 28% of the sample provided care, which is less than in other studies mentioned in the literature review. The conclusion, that "The findings demonstrate that older adults are resources to the community, mostly towards family members and therefore, should be valued as such" would appear to overstate the findings.

3. The information in Table 1 is repeated in Table 2, thus Table 1 is not necessary.

4. Table 2 has several typos (missing parentheses, stray decimal points) -- please correct.

5. Table 4 has rows that do not line up.

6. Please attend more closely to editing the manuscript. Reviewer 2 points out some grammatical or usage errors.

Reviewers' comments:

Reviewer's Responses to Questions

**Comments to the Author**

1. If the authors have adequately addressed your comments raised in a previous round of review and you feel that this manuscript is now acceptable for publication, you may indicate that here to bypass the “Comments to the Author” section, enter your conflict of interest statement in the “Confidential to Editor” section, and submit your "Accept" recommendation.

Reviewer #1: All comments have been addressed

2. Is the manuscript technically sound, and do the data support the conclusions?

Reviewer #1: Yes

3. Has the statistical analysis been performed appropriately and rigorously? 

Reviewer #1: Yes

4. Have the authors made all data underlying the findings in their manuscript fully available?

Reviewer #1: Yes

5. Is the manuscript presented in an intelligible fashion and written in standard English?

Reviewer #1: No

6. Review Comments to the Author

Reviewer #1: (No Response)

7. PLOS authors have the option to publish the peer review history of their article (what does this mean?). If published, this will include your full peer review and any attached files.

Reviewer #1: No

---

## [Author Response · Author response to Decision Letter 1]

4 Aug 2020

Dear Dr Ellen Idler, 

We are very grateful for the privilege given us to revise the manuscript. In this letter, we have responded to the reviewers’ and your comments. 

Response to reviewers 

Response to reviewers: “Correlates of older adult inpatients’ personal care provision to people with functional difficulties in Ghana”. Manuscript ID: PONE-D-19-28535

The author has addressed most of the suggestions for revision made in the two reviews. There are a few remaining issues, however, that can be addressed in a Minor Revision.

Authors appreciate the additional Editor’s comments and acknowledgement on the authors side of addressing the comments. We have therefore responded to the reviews comments thoroughly. 

1. Please reframe the research question in the Abstract and elsewhere. As stated "The purpose of the research is to determine the extent of care provision and particularly examine the socio-demographic correlates of personal care provision by older adults in Ghana" -- this research question is posed as descriptive of the extent of personal care provision by ALL older adults in Ghana (the population-level) but the data come from a highly selected sample of patients, from which no generalizations can be made to any larger group.

Thank you for raising this important comments. We have modified in the “abstract” that “The purpose of the research was to examine the personal care provision and socio-demographic correlates of personal care provision by older adults inpatients in Ghana”. This is relevant because the participants were recruited from a hospital setting. They gave their account of how they provide care before they became hospitalised. These same change has been reflected in the Title.

2. Please provide a more realistic assessment of the extent of care provision. Only 28% of the sample provided care, which is less than in other studies mentioned in the literature review. The conclusion, that "The findings demonstrate that older adults are resources to the community, mostly towards family members and therefore, should be valued as such" would appear to overstate the findings.

Thanks to the reviewer for identifying this important issue. The reviewer should be aware that we measured the extent of care provision by using three questions (Number of people you regularly care, Often assistance is provided, and Number of times spent on care –see Table 3). These three questions were asked to explore participant’s initial response to the dependent variable question “Do you regularly provide care or assistance (e.g. bathing, transport) to any other person because of their long-term illness, disability or frailty before you were hospitalised?” Authors have also altered this by providing in the “conclusion” that “Though a small proportion of older adults receiving health care from the hospital setting provide personal care, the findings in this study demonstrate that older adults in the community may be resourceful to the community and their family members due to their relatively better health. Moreover, the study’s findings draw attention to the need to provide financial and health support to older adults receiving healthcare who do not provide personal care to enable them be productive.”

Moreover, the authors have specified the nature of participants as “older adults inpatients” to differentiate participants from the general population. 

3. The information in Table 1 is repeated in Table 2, thus Table 1 is not necessary.

Thank you for this suggestion. Table 1 specifies the percentages of the individual categories the respective variables. However, in Table 2 it specifies the row total of the variables in relation to personal care. These are two different tables and due to this, the authors think that both Tables should be maintained.

4. Table 2 has several typos (missing parentheses, stray decimal points) -- please correct.

Thanks for this emphasis. Authors have addressed all errors in Table 2. 

5. Table 4 has rows that do not line up.

Thanks to the reviewer for drawing attention to this issue. I have corrected the problem in the Table 4.

6. Please attend more closely to editing the manuscript. Reviewer 2 points out some grammatical or usage errors.

Thanks very much. We have edited the manuscript for grammatical errors. Thanks

---

## [Decision Letter · Decision Letter 2]

24 Aug 2020

Correlates of older adult inpatients’ personal care provision to people with functional difficulties in Ghana

PONE-D-19-28535R2

Dear Dr. Awuviry-Newton,

We’re pleased to inform you that your manuscript has been judged scientifically suitable for publication and will be formally accepted for publication once it meets all outstanding technical requirements.

Kind regards,

Ellen L. Idler

Academic Editor

PLOS ONE

Additional Editor Comments (optional):

Reviewers' comments:

Reviewer's Responses to Questions

**Comments to the Author**

1. If the authors have adequately addressed your comments raised in a previous round of review and you feel that this manuscript is now acceptable for publication, you may indicate that here to bypass the “Comments to the Author” section, enter your conflict of interest statement in the “Confidential to Editor” section, and submit your "Accept" recommendation.

Reviewer #1: All comments have been addressed

2. Is the manuscript technically sound, and do the data support the conclusions?

Reviewer #1: Yes

3. Has the statistical analysis been performed appropriately and rigorously? 

Reviewer #1: Yes

4. Have the authors made all data underlying the findings in their manuscript fully available?

Reviewer #1: Yes

5. Is the manuscript presented in an intelligible fashion and written in standard English?

Reviewer #1: Yes

6. Review Comments to the Author

Reviewer #1: (No Response)

7. PLOS authors have the option to publish the peer review history of their article (what does this mean?). If published, this will include your full peer review and any attached files.

Reviewer #1: No

---

## [Editor Report · Acceptance letter]

24 Sep 2020

PONE-D-19-28535R2 

Correlates of older adult inpatients’ personal care provision to people with functional difficulties in Ghana 

Dear Dr. Awuviry-Newton:

I'm pleased to inform you that your manuscript has been deemed suitable for publication in PLOS ONE. Congratulations! Your manuscript is now with our production department. 

Kind regards, 

on behalf of

Professor Ellen L. Idler 

Academic Editor

PLOS ONE